# COVID-19 Mortality in Patients Aged 80 and over Residing in Nursing Homes—Six Pandemic Waves: OCTA-COVID Study

**DOI:** 10.3390/ijerph191912019

**Published:** 2022-09-22

**Authors:** Claudia Ruiz-Huerta, Marcelle V. Canto, Carmen Ruiz, Ildefonso González, Isabel Lozano-Montoya, Maribel Quezada-Feijoo, Francisco J. Gómez-Pavón

**Affiliations:** 1Department of Preventive Medicine, Hospital Central de la Cruz Roja, 28003 Madrid, Spain; 2Faculty of Medicine, “Alfonso X el Sabio” University, 28691 Madrid, Spain; 3Department of Geriatric Medicine, Hospital Central de la Cruz Roja, 28003 Madrid, Spain; 4Department of Cardiology, Hospital Central de la Cruz Roja, 28003 Madrid, Spain

**Keywords:** elderly, nursing homes, COVID-19, hospital, mortality

## Abstract

During the first COVID-19 pandemic wave in Spain, 50% of deaths occurred in nursing homes, making it necessary for some hospitals to support these facilities with the care of infected patients. This study compares origin, characteristics, and mortality of patients admitted with COVID-19 during six pandemic waves in the Hospital Central de la Cruz Roja in Madrid. It is a retrospective observational study of patients ≥80 years old, admitted with an acute SARS-CoV-2 infection, with a total of 546 patients included, whose final outcome was death or discharge. During the first wave, those from nursing homes had a higher risk of death than those from home; during the two successive waves, the risk was higher for those from home; and in the last two waves, the risk equalized and decreased exponentially in both groups. Men had 72% higher risk of death than women. For each year of age, the risk increased by 4% (*p* = 0.036). For each Charlson index point, the risk increased by 14% (*p* = 0.019). Individuals in nursing homes, despite being older with higher comorbidity, did not show a higher overall lethality. The mortality decreased progressively in each successive wave due to high vaccination rates and COVID-19 control measures in this population.

## 1. Introduction

Following the declaration of the state of alarm on 15 March 2020, Madrid became the epicenter of the COVID-19 pandemic in Spain [1], accounting for the highest incidence rate in the country (907.75 per 100,000 inhabitants), as well as the highest lethality (12.7%) and ICU bed occupancy rate [1]. The elderly population suffered the greatest impact from the pandemic, with a 21.8% lethality in Spain as of 29 May 2020 in patients over 80 years of age (12,834 deaths) [2], with 10,121 confirmed deaths being registered only in Madrid [3]. High mortality rate in this age group has been observed constantly throughout the world, with lethalities ranging from 14.8%, referred in the first publications from China [4,5], to 26.4% mentioned in the Bonanad et al. meta-analysis [6], or 55.4% reported in the first Spanish hospital cohort published in this country [7].

However, even though the majority of admissions and deaths cases have been concentrated in older people, it is not possible to establish an age threshold above which the risk is higher, due to the coexistence of additional factors that contribute to this increased risk, highlighting comorbidities and institutionalization [8,9]. Thus, as of 23 June 2020, the impact of COVID-19 on elderly people residing permanently in nursing homes was 19,533 deaths nationally, which represented 68.1% of the confirmed deaths due to COVID-19 up to that time [9]. It is estimated that, of the total number of deaths during the first pandemic wave in Spain, 47–50% occurred in nursing home facilities [10].

Successive pandemic waves, lasting a period of two years, continued to generate new infections, causing a high number of deaths due to COVID-19 in vulnerable populations (registering 105,642 deaths in Spain as of 15 May 2022) [11]. The incidence was particularly high among those over 80 years of age (67,865 deaths in Spain as of 17 May 2022 in this population group) [12].

The arrival of an effective vaccine against the SARS-CoV-2 infection marked a turning point in the mortality of this disease [13,14,15]. Vaccination of vulnerable populations in Spain began in December 2020 and, since then, mortality rates have started to decrease significantly [16,17].

In November 2021, the emergence of the new Omicron variant generated an increased number of cases of infection [18]. However, given the high vaccination rate at that period in our country and the characteristics of the new variant, the sixth pandemic wave did not have a great impact on elderly patients’ mortality [19,20].

This research aims to compare, taking into account living arrangements (living at home or permanently in a nursing home), the characteristics and mortality of older patients admitted with COVID-19 in the Geriatric Unit of the Hospital Central de la Cruz Roja (HCCR) in Madrid during the six pandemic waves period.

## 2. Materials and Methods

### 2.1. Study Design and Participants

We conducted a retrospective observational longitudinal cohort study including 546 patients aged 80 years or older. We used data from the hospital discharge records database (CMBD) of patients admitted to HCCR between 10 March 2020 and 10 February 2022, with a diagnosis of acute SARS-CoV-2 incident infection. Validated diagnosis codes associated with COVID-19 according to the International Classification of Diseases (ICD-10-ES), third edition, January 2020, were used during the study period [21]. Records with ICD-10 code B34.2 (COVID-19 infection, unspecified), B97.2 (COVID-19 as the cause of diseases classified elsewhere), B97.21 (COVID-19 associated with SARS as the cause of diseases classified elsewhere), B97.29 (other COVID-19 as the cause of diseases classified elsewhere), U07.1 (COVID-19), and U07.2 (COVID-19, virus no identified) were analyzed as possible COVID-19 confirmed cases. Records with codes J12.89 (other types of viral pneumonia), J12.81 (pneumonia, associated with SARS), and J18.9 (pneumonia, unspecified microorganism) were also analyzed as COVID-19 probable cases and at a later stage, they were confirmed or excluded using the Department of Preventive Medicine record of confirmed COVID-19 hospitalized patients. For each registry entry, main diagnosis (the major process that was considered the primary reason for the patient’s hospital admission) and secondary diagnoses were analyzed; secondary diagnoses might have coexisted from the beginning with the main diagnosis at the time of admission or they might have developed during the hospital stay.

The HCCR is a general medical–surgical support hospital (level II) in the Community of Madrid [22] which, since early March 2020, received patients with SARS-CoV-2 acute infection, many of whom came from nursing homes facilities. Our Geriatric Department has teams providing hospital care directly to nursing homes (Geriatrics Consultation Liaison) that travel, periodically, to those facilities, covering nearly 4000 individuals permanently residing in nursing homes in the north and northwest areas of Madrid. In the first pandemic wave, the HCCR provided support for COVID-19 patients from all areas of the Community of Madrid, although preference was given to patients from its main area of coverage, but after the successive pandemic waves, care was provided only for nursing homes in the areas to which it habitually supplies coverage.

During this study period, six pandemic waves occurred in the Community of Madrid, and 546 octogenarian or older patients with a SARS-CoV-2 incident infection that met the inclusion criteria were admitted to the HCCR. These criteria were as follows: patients equal to or over 80 years of age, with a recent confirmed diagnosis of COVID-19 by a positive diagnostic test for acute infection (RT-PCR or antigen test) or with a probable diagnosis of COVID-19 with a high clinical–radiological suspicion but with a negative test at the moment of hospitalization, admitted to our hospital in the Geriatric Unit. These diagnosis data were obtained from the Preventive Medicine Department, which is the service that has an updated list of all COVID-19 patients in the hospital, as well as their classification depending on the time of diagnosis and the stage of the infection. Patients who refused to participate in the study were excluded and their data were anonymized. Additionally, patients readmitted with prolonged COVID-19 diagnosis, resolved infection, or with a diagnosis of sequalae due to COVID-19 and posterior to their primary infection were excluded.

All patients were included in the cohort at the same time after their demise or hospital discharge based on data from the CMBD records. We used the date of hospitalization and the date of death or discharge as key elements to display a timeline in our cohort. The first pandemic wave included patients who were hospitalized from 10 March through 10 July 2020; the second wave, those admitted between 11 July and 8 December 2020; and the third wave, from 9 December 2020 through 10 March 2021. During the fourth wave, no admissions with a diagnosis of COVID-19 were registered at our center. The fifth wave covered the period between 1 July and 31 October 2021 and, finally, the sixth wave covered the dates between 1 November 2021 and 10 February 2022.

### 2.2. Variables

Demographic variables were collected from the medical records of each admitted patient, such as sex, age, and living arrangements (living at home or permanently at a nursing home), in addition to clinical variables, such as diagnosis of COVID-19 status (confirmed or probable), most common comorbidities associated with SARS-CoV-2 infection according to the literature (chronic obstructive pulmonary disease (COPD), diabetes mellitus, cardiovascular disease, arterial hypertension, chronic kidney disease, chronic liver disease, neoplasms, dementia, or neurological sequelae) and age-adjusted Charlson index (calculated taking into account the associated comorbidities at admission), average length of hospital stay, and hospital discharge status (death or cure). Data were entered manually and stored routinely in an Excel database.

### 2.3. Statistical Analysis

Due to the characteristics of the study, the required sample size was not calculated and it was considered equal to the total number of patients admitted during the study period. To synthesize the analysis data, frequencies and percentages were used for categorical variables, and means with their standard deviations (SD) were used for continuous variables.

A chi-squared test (X^2^ test) or a Fisher’s exact test with a 95% confidence interval was used to evaluate the difference in the distribution of qualitative variables. For quantitative and categorical variables, a Student’s *t*-test was used for continuous variables and an analysis of variance (ANOVA) test for variables containing three or more categories.

A logistic regression model was used for multivariate analysis to measure the variables associated with COVID-19 mortality in our patients. In the multivariate analysis, the following variables were considered: (a) age and sex since are biologically relevant; (b) all other variables that result as significantly associated at the univariate analysis; (c) other variables considered relevant in the previous bibliographic consultation. Data were shown as adjusted odds ratios (OR) with a 95% confidence interval (CI 95%) and their corresponding *p*-values. Data analysis was performed with the Excel program, as well as with the statistical software programs SPSS^®^ version 26.0 (IBM) and STATA^®^ version 15.0.

## 3. Results

### 3.1. Descriptive Analysis, Temporal Evolution, and Univariate Analysis

From March 2020 to February 2022, a total of 546 cases admitted to HCCR with COVID-19 were registered in the Geriatric Unit, of which 421 (77.1%) corresponded to confirmed cases with a positive acute infection test prior to or during admission, while the remaining were diagnosed based on a high clinical–radiological suspicion; however, they presented with a negative test result during the hospitalization. Most of the patients were admitted during the first pandemic wave (67.8%), with a progressive decrease of the incidence in subsequent waves.

During the six pandemic waves, more women than men were admitted (65% vs. 35%), and more than half of our patients were residing permanently in nursing homes prior to their admission (61.5%). Ages ranged from 80 to 106 years, with a total age mean of 90 years at admission.

Considering only those patients who were residing permanently in a nursing homes prior to admission, 237 were women, with a mean age of 90 years (those living at home had a mean age of 88 years), a mean Charlson comorbidity index of 7.2 (±1.8) points, and a mean length of stay of 13 (±10) days (Table 1). No relevant significant differences were found when comparing the profiles of the patients only coming from a permanent nursing home during the six pandemic waves periods. A slight difference was noted in terms of mortality in these groups of patients, taking into account the pandemic wave during which patients were admitted, with a decreasing trend in subsequent waves, except during the last one, where a new increase in lethality was observed in this population (39.4% in the first, 25% in the second, 20% in the third, and 17.6% in the fifth, in comparison to 23.1% in the last wave).

When patients were compared according to their living arrangements, significant differences were found in terms of sex during the second wave, as well as in the total population, with greater frequency in women (*p* = 0.035 and *p* = 0.001, respectively), and in age, with patients residing permanently in nursing homes being older in age. This finding remained constant during all six waves (*p* = 0.000) (Table 1). Differences were also found when comparing the prevalence of neurological disease, dementia, and neurological sequelae, with the first two being more frequent in the population from nursing homes (*p* = 0.012 and *p* = 0.000, respectively). On the contrary, neurological sequelae were seen more frequently in the population living at home prior to their hospitalization (*p* = 0.000). These findings were observed in all waves except the second one (Table 2).

During the first pandemic wave, differences in lethality were found when comparing both population groups, but these were not statistically significant. In the second wave, however, lethality was shown to be more than double in patients living at home compared to those from permanent nursing homes prior to admission (52.6% vs. 25%, *p* = 0.032). This total lethality during this second wave was the highest one compared to the rest of the periods (41.9%) (Table 2). However, no significant differences were found when comparing the overall lethality in both groups of patients according to their living arrangements (*p* = 0.721). Figure 1 shows the distribution of admissions and deceased patients admitted in HCCR, highlighting the number of deaths concerning those who particularly came from permanent nursing homes.

Overall lethality increased with age, resulting in 31.3% in octogenarians, 37.2% in nonagenarians, and 40% in those patients over 100 years old. Men had higher risk of death compared to women, and this difference was found to be statistically significant (41.9% vs. 30.1%, *p* = 0.006). When morbidity was taken into account, differences in mortality were observed in patients with diabetes, representing a total of 42.3% of those who passed away (*p* = 0.012) (Table 3). No significant differences were found in mortality or the remainder of the comorbidities studied.

### 3.2. Multivariate Analysis

We performed a multivariate analysis of the risk factors associated with mortality in patients admitted with COVID-19 in HCCR during the study period and found significant differences in terms of sex, age, and Charlson comorbidity index, as well as differences during the fifth wave when compared to the first pandemic wave (Table 4).

In the overall adjusted analysis, we found that men had a 72% higher mortality risk than women (*p* = 0.006). It was observed that, for each year of age, the risk of death increased by 4% (*p* = 0.036) and having diabetes mellitus increased the risk of dying by 42% (*p* = 0.115). For each one-point increase in the Charlson comorbidity index, the risk of death increased by 14% (*p* = 0.019). As for the analysis by pandemic waves, it was observed that, during the second wave, the risk of death was 19% higher than during the first wave period (*p* = 0.551). On the contrary, during the third, fifth, and sixth waves, the risk of death decreased by 51%, 66%, and 45%, respectively, compared to the risk of having been infected and admitted during the first pandemic wave.

In the crude analysis, it was observed that residing in a nursing home prior to the hospital admission entailed a 7% higher risk of death than living at home (*p* = 0.721), but this risk was null in the adjusted analysis.

When those patients residing at nursing homes were analyzed separately, it was observed that men had a 33% higher risk of death than women (*p* = 0.261), and that, for each year of age, this risk increased by 2% (*p* = 0.508). With regard to the pandemic waves during which they were admitted to our hospital, those who entered during the second (48%), third (63%), fifth (66%), and sixth waves (51%) had a decreased risk of mortality compared to those who were admitted during the first pandemic wave. However, no significant differences were found with any of the variables analyzed in this population group (Figure 2).

## 4. Discussion

This study reports the experience gathered from a support hospital in the Community of Madrid (CM) with patients over 80 years of age with COVID-19, comparing patient characteristics and mortality according to their living arrangements (living at home or in a permanent nursing home) over six pandemic waves. The study provides important data: during the first pandemic wave, patients over 80 years of age residing in nursing homes had a higher risk of dying from COVID-19 during hospital admission than those who lived in their homes; during the second and third waves, the risk was higher for those living at home, and during the fifth and sixth waves, the risk was equal in both population groups.

The higher mortality of men than women detected in the first wave, as shown in most studies, could be explained by differences regarding innate and adaptive immunity linked to sex, which may cause a different inflammatory response to COVID-19 [22,23] and the distinct sex distribution of associated comorbidities according to lifestyle. Age and comorbidities, such as dependence, frailty, dementia, or a high number of chronic diseases [24,25,26,27,28,29,30,31,32], are defining characteristics of the geriatric patient and determinants of the severity of any other ensuing disease, and even more so of COVID-19 [33].

The incidence rate of SARS-CoV-2 infection is one of the most explanatory parameters of mortality, especially in the elderly population, where mortality can be even higher. According to the study by Medeiros Figueroa et al. of the CIBER Center of Epidemiology and Public Health in Spain [1], the incidence rate in the CM was close to one thousand cases per 100,000 inhabitants; however, the real rate must have been higher, especially in nursing homes, due to the difficulty in performing diagnostic tests at that time [34]. Thus, the excess in mortality of this population could be related to the high incidence of infection in these facilities, as they are institutions of mostly shared living spaces [35].

This was a cohort with a large number of patients admitted to our hospital from permanent nursing homes; almost 60% of our admissions were patients from these facilities, with a COVID-19 lethality in the initial phases of the pandemic of 40.8%, which is similar to that reported by other publications, but with lower percentages than in other studies involving patients from nursing homes [25,36]. The higher risk of death associated with this population group during the first wave could also be related to the intrinsic characteristics of institutionalized patients with greater clinical frailty [37], higher age, functional dependence, and comorbidities, especially neurological diseases and dementia, which, in turn, favor a higher risk of developing delirium during hospital stay, an especially decisive risk factor for mortality in the elderly [38].

One of the first Spanish studies published by the Hospital de la Paz scientific group in Madrid [7], with a cohort of 2226 admitted patients diagnosed with COVID-19, 31.9% of whom came from permanent nursing homes, showed a lethality of 52.9% in those over 80 years of age, rising to 63.3% in those over 90 years of age. A multicenter study of Spanish hospitals [24], mainly situated in Madrid, showed data from a sample of 4035 COVID-19 patients, with a lethality of 54.9% in this population group.

Our experience in the first wave with 370 admitted patients showed a lethality of 36.3% in those over 80 years of age and 43.9% in those over 90 years of age, which is more in line with a multicenter European study, where a larger number of British hospitals with geriatric-preferential units were included [26]. This study involved 1564 patients and showed a lethality of 37.3% in those over 80 years of age. These data, with lower mortality than the ones observed in similar studies, could be related to an advanced experience in managing geriatric patients in our hospital, with comprehensive geriatric assessment due to the existence of a working tool that allows early recognition and treatment of atypical complications and geriatric syndromes associated with admission for COVID-19 in frail elderly patients [33]. Another factor that may have influenced these data is that patients who are candidates for intensive care therapy are transferred to referral hospitals with an intensive care unit (ICU), although, contrary to this assessment, the profile of patients admitted to our center are usually geriatric patients with high dependency and frailty, most of whom are not candidates for ICU and are therefore at high risk of dying.

In the second and third pandemic waves, patients living at home had a higher risk of death during hospital admission due to COVID-19 than those residing permanently in nursing homes. This finding may have been related to the management of the elderly in nursing homes by the hospital due to a coordination between geriatric liaison teams, created in the first wave, and primary care nursing home supervision units, with direct communication with the public health department, as well as a greater facilitation of diagnostic tests in those centers. However, regarding patients living at home, in addition to the fear of the elderly of visiting hospitals because of the potential risk of contracting SARS-CoV-2 infection, there is an overloaded primary care that is overburdened with great difficulties in the management of the elderly directly at home. These two factors could explain why the elderly patients living at home will visit the hospital in more severe and advanced clinical situations, probably causing a higher in-hospital mortality in this population group [39]. The excessive mortality in nursing homes during the first wave, that affected the most vulnerable patients, should also be taken into account.

In December 2020, the vaccination campaign against SARS-CoV-2 infection was initiated in nursing homes in Spain, with the administration of the first dose of the BNT162b2 vaccine and, sequentially, a second dose in January 2021 [10,16]. As a consequence of the vaccination campaign, a reduction in the number of deaths caused by COVID-19 among the most vulnerable was reported [17].

During the fourth pandemic wave, our hospital did not record patients admitted with COVID-19, as hardly any cases were recorded in institutionalized patients, due to both high vaccination coverage [40] and the likelihood to have recently passed the infection in previous waves [41]. This fourth wave was caused mostly by the Omicron variant, which turned out to be less lethal [42,43] and was also in a period when booster doses of the SARS-CoV-2 vaccine were already being administered [44,45].

During the fifth wave, vaccination rates against SARS-CoV-2 in the octogenarian population were close to 100%, with the full vaccination schedule completed [42]. In this pandemic wave, the lowest lethality ever was recorded amongst our patients: ranking from 35.4% during the first wave, dropping down to 17.2% during the fifth wave. Moreover, in the latter, no differences in mortality were found between patients living at home and those permanently residing in nursing homes. The high vaccination coverage achieved at that time was a determining factor in the reduction of mortality in the most vulnerable patients [16,17], regardless of their living arrangements prior to hospital admission. The same could be seen during the sixth wave [20], which was also caused mostly by the Omicron variant. Again, during this period, no differences in mortality were observed between patients admitted from home and those residing permanently in nursing homes.

It should also be taken into consideration the increased risk that posed the likelihood of contracting the infection in a nursing home, which is much higher due to the characteristics of these facilities. For this reason, although the case fatality could be the same, the risk of contracting COVID-19, and subsequently dying due to this disease, could be much different in the population.

Another point to consider regarding the factors that could have affected the mortality rate is the role that the different infection treatments or supportive measures applied to patients with COVID-19 throughout the pandemic might have played, which could have been very different in the first and last waves, or, for example, the differences between those patients who received monotherapy or multiple therapies [46]. Scientific advances brought to light new drugs to treat COVID-19, which led to changes in the cure of the disease. In addition, gender differences also need to be taken into account regarding the role of therapy, as described by Spini et al. in a study of elderly people in nursing homes in Italy, where gender-related differences in the performance of drugs for the treatment of COVID-19 were found [47].

## 5. Strengths and Limitations

The strength of this study lies in the sequential collection of information from patients over 80 years of age admitted with COVID-19 and their follow-up during their hospital admission, with a high proportion of them residing in nursing homes prior to the admission. Among the limitations, we would like to mention the noninclusion of important geriatric variables, such as the degree of frailty or dependence, clearly associated with mortality in octogenarian patients, although in our study this is reflected by the greater comorbidity due to dementia in the nursing home population. On the other hand, the results only include cases of COVID-19 admitted to our hospital, so they do not reflect asymptomatic infections that required medical intervention but were managed at home or directly in the nursing home facilities. The profile of hospital admissions could be different in the distinctive waves, due to the implementation of various restrictions and infection transmission control measures at the national level, and the sample sizes of some waves were considered small, therefore, the estimates may be less precise and there may be less probability of finding significant differences between groups. In addition, the study could not differentiate between patients admitted strictly for COVID-19 and asymptomatic patients hospitalized for other reasons who tested positive by chance during their admission. In addition, the pathogenicity of the variants in each wave also may have played a role this.

It should be also noted as a limitation that neither the treatments received nor their vaccination status were available for this study. Finally, it is worth noting the absence of follow-up after discharge, so that only in-hospital mortality can be estimated, as some of these patients might have been readmitted at a later date.

## 6. Conclusions

Octogenarian and even older patients from nursing homes hospitalized with COVID-19 in our Geriatric Unit, despite being older and having more comorbidities than those that lived at homes prior to the admission, did not present a higher overall risk of death. During the first wave, a higher mortality was recorded in this population group, but it decreased progressively in each of the successive waves due to the high vaccination rates in this crowd, the difference in pathogenicity of the variants of each wave, and the COVID-19 coordination and control measures developed in the environments of the nursing home facilities.

## Figures and Tables

**Figure 1 ijerph-19-12019-f001:**
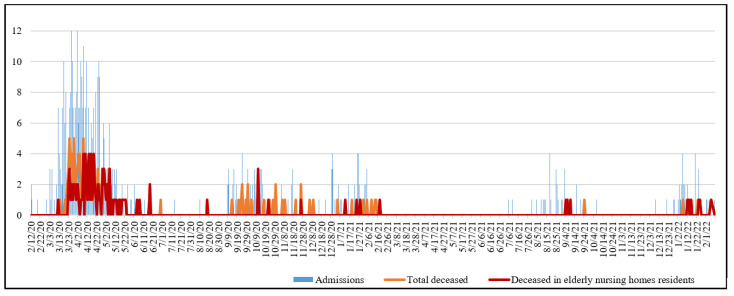
Evolution of hospital admissions and deceased patients because of COVID-19 during six pandemic waves. HCCR.

**Figure 2 ijerph-19-12019-f002:**
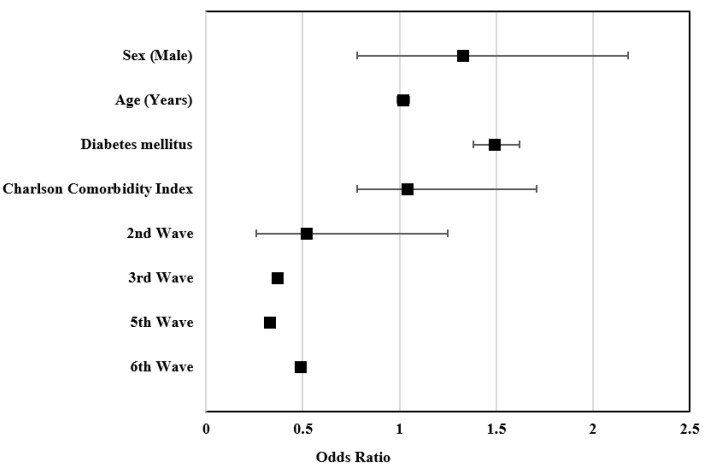
Risk factors associated with mortality from COVID-19 in patients from nursing homes: multivariate analysis.

**Table 1 ijerph-19-12019-t001:** Clinical and sociodemographic factors of COVID-19 patients in HCCR, differences between living arrangement (living at home vs. nursing home).

		Living at Home	Nursing Home	Total	*p*-Value
**Factor, n (%)**		(n = 210)	(n = 336)	(n = 546)	
**Sex**	Female	118 (33.2)	237 (66.8)	355 (65)	0.001
	Male	92 (48.2)	99 (51.8)	191 (35)	0.001
**Mean Age (±SD); Years**		88.2 (±4.6)	90.3 (±5.1)	89.5 (±5)	0.000
**Age Group**	80–89 years old	130 (47.3)	145 (52.7)	275 (50.4)	0.000
	≥90 years old	80 (29.5)	191 (70.5)	271 (49.6)	0.000
**COVID-19 Diagnosis**	Confirmed	178 (42.3)	243 (57.7)	421 (77.1)	0.001
	Probable	32 (25.6)	93 (74.4)	125 (22.9)	0.001
**Risk Factors**	Chronic respiratory disease	33 (45.8)	39 (54.2)	72 (13.2)	0.168
	Diabetes mellitus	61 (39.1)	95 (60.9)	156 (28.6)	0.846
	Arterial hypertension	93 (37.3)	156 (62.7)	249 (45.6)	0.625
	Cerebrovascular disease	126 (41.9)	175 (58.1)	301 (55.1)	0.070
	Chronic renal insufficiency	46 (38)	75 (62)	121 (22.2)	0.909
	Chronic hepatic disease	2 (66.7)	1 (33.3)	3 (0.5)	0.314
	Neurological disorders	24 (26.7)	66 (73.3)	90 (16.5)	0.012
	Neoplastic disease	29 (38.2)	47 (61.8)	76 (13.9)	0.953
	Dementia	39 (25.5)	114 (74.5)	153 (28)	0.000
	Neurological sequelae	27 (65.9)	14 (34.1)	41 (7.5)	0.000
**Charlson Comorbidity Index; Mean (±SD)**		6.8 (±2)	7.2 (±1.8)	7.1 (±1.9)	0.021
**Charlson Index**	4 points	24 (68.6)	11 (31.4)	35 (6.4)	0.020
	5 points	36 (40.9)	52 (59.1)	88 (16.1)	0.020
	6 points	35 (33.7)	69 (66.3)	104 (19)	0.020
	7 points	47 (39.2)	73 (60.8)	120 (22)	0.020
	≥8 points	68 (34.2)	131 (65.8)	199 (36.4)	0.020
**Length of Hospital Stay; Mean (±SD); Days**		17.6 (±15.1)	13.3 (±10.2)	14.9 (±12.5)	0.000
**Lethality**		70 (33.3)	117 (34.8)	187 (34.2)	0.721

SD: standard deviation.

**Table 2 ijerph-19-12019-t002:** Clinical and sociodemographic factors of patients infected by SARS-CoV-2, differences according to living arrangement and COVID-19 pandemic waves in HCCR.

		1st Wave		2nd Wave		3rd Wave		5th Wave		6th Wave	
		Living at Home	Nursing Home	Living at Home	Nursing Home	Living at Home	Nursing Home	Living at Home	Nursing Home	Living at Home	Nursing Home
**Factor, n (%)**		(n = 121)	(n = 249)	(n = 38)	(n = 24)	(n = 26)	(n = 20)	(n = 12)	(n = 17)	(n = 13)	(n = 26)
**Sex**	Female	72 (29.6)	171 (70.4)	20 (51.3)	19 (48.7) *	11 (44)	14 (56)	7 (36.8)	12 (63.2)	8 (27.6)	21 (72.4)
	Male	49 (38.6)	78 (61.4)	18 (78.3)	5 (21.7) *	15 (71.4)	6 (28.6)	5 (50)	5 (50)	5 (50)	5 (50)
**Mean Age (±SD); Years**		88.1 (±4.6)	90.1 (±5.1) *	87.9 (±5)	92.7 (±4.9) *	88.8 (±3.9)	89.9 (±4.5)	88.3 (±5)	90 (±5.2)	88.2 (±4.3)	89.8 (±5.2)
**Age Group**	80–89 years old	77 (41.2)	110 (58.8) *	23 (82.1)	5 (17.9) *	15 (65.2)	8 (34.8)	8 (50)	8 (50)	7 (33.3)	14 (66.7)
	≥90 years old	44 (24)	139 (76) *	15 (44.1)	19 (55.9) *	11 (47.8)	12 (52.2)	4 (30.8)	9 (69.2)	6 (33.3)	12 (66.7)
**Risk Factors**	Chronic respiratory disease	18 (40)	27 (60)	6 (75)	2 (25)	5 (83.3)	1 (16.7)	1 (25)	3 (75)	3 (33.3)	6 (66.7)
	Diabetes mellitus	32 (30.8)	72 (69.2)	16 (72.7)	6 (27.3)	5 (41.7)	7 (58.3)	5 (55.6)	4 (44.4)	3 (33.3)	6 (66.7)
	Arterial hypertension	48 (31.4)	105 (68.6)	20 (58.8)	14 (41.2)	7 (43.8)	9 (56.3)	5 (41.7)	7 (58.3)	13 (38.2)	21 (61.8)
	Neurological disorders	15 (23.1)	50 (76.9)	4 (66.7)	2 (33.3)	3 (30)	7 (70)	1 (25)	3 (75)	1 (20)	4 (80)
	Dementia	26 (25)	78 (75) *	4 (44.4)	5 (55.6)	5 (35.7)	9 (64.3)	2 (40)	3 (60)	2 (9.5)	19 (90.5) *
**Charlson Comorbidity Index; Mean (±SD)**		6.6 (±1.9)	7.2 (±1.8) *	7.2 (±2.1)	6.8 (±1.7)	7.2 (±2.1)	7.3 (±2.1)	6.9 (±1.5)	7.3 (±1.5)	6.5 (±1.3)	7.1 (±1.8)
**Length of Hospital Stay; Mean (±SD); Days**		16.4 (±14.8)	12.8 (±10) *	17 (±13)	13.7 (±10.1)	20.2 (±14.2)	16.5 (±9.4)	25 (±25.6)	18.2 (±14.5)	18.6 (±11.5)	11.6 (±9.4) *
**Lethality**		38 (31.4)	98 (39.4)	20 (52.6)	6 (25) *	7 (26.9)	4 (20)	2 (16.7)	3 (17.6)	3 (23.1)	6 (23.1)

* Statistical significance (*p* < 0.05). SD: standard deviation.

**Table 3 ijerph-19-12019-t003:** Clinical and sociodemographic factors among patients who survived and died, infected by SARS-CoV-2 in HCCR.

Factor, n (%)		Survivors	Deceased	Total	*p*-Value
		(n = 359)	(n = 187)	(n = 546)	
**Sex**	Female	248 (69.9)	107 (30.1)	355 (65)	0.006
	Male	111 (58.1)	80 (41.9)	191 (35)	0.006
**Mean Age (±SD); Years**		89.1 (±5)	90.2 (±4.9)	89.5 (±5)	0.012
**Age Group**	80–89 years old	189 (68.7)	86 (31.3)	275 (50.4)	0.331
	≥90 years old	170 (62.7)	101 (37.3)	271 (49.6)	0.331
**Living Arrangement**	Living at home	140 (66.7)	70 (33.3)	210 (38.5)	0.721
	Nursing home	219 (65.2)	117 (34.8)	336 (61.5)	0.721
**COVID-19 Diagnosis**	Confirmed	279 (66.3)	142 (33.7)	421 (77.1)	0.639
	Probable	80 (64)	45 (36)	125 (22.9)	0.639
**Pandemic Waves**	1st wave	234 (63.2)	136 (36.8)	370 (67.8)	0.030
	2nd wave	36 (58.1)	26 (41.9)	62 (11.4)	0.030
	3rd wave	35 (76.1)	11 (23.9)	46 (8.4)	0.030
	5th wave	24 (82.8)	5 (17.2)	29 (5.3)	0.030
	6th wave	30 (76.9)	9 (23.1)	39 (7.1)	0.030
**Risk Factors**	Chronic respiratory disease	44 (61.1)	28 (38.9)	72 (13.2)	0.373
	Diabetes mellitus	90 (57.7)	66 (42.3)	156 (28.6)	0.012
	Arterial hypertension	170 (68.3)	79 (31.7)	249 (45.6)	0.255
	Cerebrovascular disease	188 (62.5)	113 (37.5)	301 (55.1)	0.072
	Chronic renal insufficiency	71 (58.7)	50 (41.3)	121 (22.2)	0.063
	Chronic hepatic disease	2 (66.7)	1 (33.3)	3 (0.5)	0.973
	Neurological disorders	57 (63.3)	33 (36.7)	90 (16.5)	0.597
	Neoplastic disease	45 (59.2)	31 (40.8)	76 (13.9)	0.195
	Dementia	105 (68.6)	48 (31.4)	153 (28)	0.377
	Neurological sequelae	24 (58.5)	17 (41.5)	41 (7.5)	0.311
**Charlson Comorbidity Index; Mean (±SD)**		6.8 (±1.8)	7.5 (±1.8)	7.1 (±1.9)	0.000
**Charlson Index**	4 points	29 (82.9)	6 (17.1)	35 (6.4)	0.008
	5 points	69 (78.4)	19 (21.6)	88 (16.1)	0.008
	6 points	66 (63.5)	38 (36.5)	104 (19)	0.008
	7 points	78 (65)	42 (35)	120 (22)	0.008
	≥8 points	117 (58.8)	82 (41.2)	199 (36.4)	0.008
**Length of hospital stay; Mean (±SD); Days**		16.8 (±12.6)	11.3 (±11.5)	14.9 (±12.5)	0.000

SD: standard deviation.

**Table 4 ijerph-19-12019-t004:** Risk factors associated with COVID-19 mortality. Multivariate analysis.

Variables	OR (CI 95%)	*p*-Value	Adjusted OR(CI 95%)	*p*-Value
**Sex (Male)**	1.67 (1.16–2.41)	0.006	1.72 (1.17–2.52)	0.006
**Age (Years)**	1.05 (1.01–1.08)	0.012	1.04 (1.00–1.08)	0.036
**Living Arrangement (Nursing Home)**	1.07 (0.74–1.54)	0.721	1.00 (0.67–1.49)	0.990
**Diabetes Mellitus**	1.63 (1.11–2.39)	0.012	1.42 (0.92–2.18)	0.115
**Charlson Comorbidity Index**	1.20 (1.09–1.32)	0.000	1.14 (1.02–1.27)	0.019
**2nd Wave**	1.24 (0.72–2.15)	0.436	1.19 (0.67–2.12)	0.551
**3rd Wave**	0.54 (0.27–1.10)	0.090	0.49 (0.23–1.02)	0.055
**5th Wave**	0.36 (0.13–0.96)	0.041	0.34 (0.13–0.93)	0.036
**6th Wave**	0.52 (0.24–1.12)	0.094	0.55 (0.25–1.22)	0.141

OR: odds ratio. CI: confidence interval.

## Data Availability

The data that support the findings of this study are available upon request from the corresponding author. The data are not publicly available due to privacy or ethical restrictions.

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
