# Peer review of "COVID-19 Mortality in Patients Aged 80 and over Residing in Nursing Homes—Six Pandemic Waves: OCTA-COVID Study"

_ijerph, 2022, doi:10.3390/ijerph191912019_

Round 1
Reviewer 1 Report (Previous Reviewer 1)
This manuscript has been improved.
This manuscript is a resubmission of an earlier submission. The following is a list of the peer review reports and author responses from that submission.
Round 1
Reviewer 1 Report
The manuscript of Claudia Ruiz-Huerta and colleagues describes a retrospective observational study of old patients with acute SARS-Cov-2 infections during 6 waves of COVID-19 pandemics. They found old patients residing in permanent home and nursing home have different risk of death among different pandemics. In addition, sex and age are identified as factors contributing to the risk of death of COVID-19 infection. The analysis and results are reasonable and sound. The limitations have been described properly. The following comments are aimed at improving the quality of the work.
Minor comments:
1. Figure 1 is not clear. Please provide the legend of orange color in figure 1.
2. Please describe why the authors determined the 6 waves or the standards on making the dates (Line 95-100).
3. The sample sizes of wave 3-6 are small (<50), which should also be documented in the limitation and when drawing conclusions among these waves.
4. The less death rate in the later pandemic waves should be related to the vaccination rates, control measures. But also, the pathogenicity of the variants of each wave also plays a role in it. The author should take this into account and discuss it properly.
Reviewer 2 Report
Dear authors,
You give interesting information COVID-19 mortality in patients over 80 residing in nursing homes. I have some comments.
Methods:
- Line 66: What CMDB stands for?
- Line 68: Please specify what registers are you talking about. Did you use medical records or administrative data? From the description it seems you used hospital discharge records (main, secondary diagnoses), however you stated that you included only patients with confirmed diagnosis of covid. I suggest you to describe better your datasource.
- Line 120: Please be explicit about the desired output of the logistic regression analysis.
- What is the date of cohort entry? Date of hospitalization for COVID?
- Do you have any information about vaccine status and drugs that these patients have received?
Results:
- I suggest authors to remove table 2 from the main manuscript and report it in the supplementary.
- Legend of figure 1 is missing. What red and orange line stand for?
- I guess that table 4 is more informative than table 3. Why did you reported both?
- If I got right figure 2 is a stratification of table 4. I suggest the authors to choose how to report the results of multivariate logistic regression (Figure o table?) I believe it could be interesting if in figure 2 there are also the results of those patients in permanent homes.
Discussion:
- Line 220-227: I guess another point needs to be added. Also the different use of drugs between men and women with COVID-19 could play a role in the mortality of such patients (PMID: 35194891, PMID: 33903671)
- If it’s impossible to retrieve vaccination status and treatment received for patients included in the cohort, please add in the limitation section.
Round 2
Reviewer 2 Report
Dear authors, you can find some additional comments to your answer.
R1: Line 120: Please be explicit about the desired output of the logistic regression analysis
A1: A logistic regression model was used to measure the impact that residency status, comorbidities, sex, age, Charlson Index and the period wave of hospitalization, have in COVID-19 mortality in our patients.
R2: Please report it in the text. It is not clear.
-----------------------------------------------
R1: What is the date of cohort entry? Date of hospitalization for COVID?
A1: The study is retrospective; therefore, all patients were included in the cohort at the same time with the CMBD records. We used the date of hospitalization and the date of death or discharge, as key dates to display a timeline in our cohort.
R2: So please report in the text that the date of cohort entry in your retrospective cohort study is the first date of hospital discharge record with a primary/secondary COVID-19 codes.
Are the included patients incident patients? It is not clear from the manuscript. Are these codes that you used been previously validated for the identification of COVID-19 patients?
Moreover, what you stated is not in line with manuscript text since you reported that “also patients with suggestive symptoms of COVID detected by the Preventive 77 Medicine Department” were added. I guess these are not patients identified from hospital discharge records.
Methods of inclusion of the patients are still not clear. I strongly suggest authors to deeply revise method section. 1) briefly describe study design, 2) describe datasources 3) Describe inclusion and exclusion criteria (how do you search incident patients? Did you include those patients with a first record?)
---------------------------------------------------
R1: Legend of figure 1 is missing. What red and orange line stand for?
A1: The legends of Figure 1 and the rest of the graphic are display in the Microsoft Word document, but are missing in the PDF format due to incompatibility of the paper disposition. We fixed it and adjusted to a vertical position.
R2: I don’t see any difference with the previous version.
----------------------------------------------------
R1: If I got right figure 2 is a stratification of table 4. I suggest the authors to choose how to report the results of multivariate logistic regression (Figure o table?) I believe it could be interesting if in figure 2 there are also the results of those patients in permanent homes.
A1: Thank you for your comment. Table 4 includes the risk factors associated with COVID-19 mortality in the entire cohort, regardless of where they usually reside, quantifying the impact, that each of the variables has on mortality. Figure 2, however, refers to only the group of patients who come from residence, in which, the impact of each of the variables is represented graphically for this population group. We considered that including this information in a figure was a more visual way to show the mortality risk factor in this vulnerable population that reside in nursing homes.
R2: My suggestion was to add also figure for those in permanent homes. Why didn't you report both stratification data but only for patients in nursing homes? All result need to be shown
----------------------------------------------------
R1: Line 220-227: I guess another point needs to be added. Also the different use of drugs between men and women with COVID-19 could play a role in the mortality of such patients (PMID: 35194891, PMID: 33903671)
A1: Thank you for your interesting point of view. It would certainly be interesting to analyze this information. However, for this study we didn´t have available the information regarding the medication used in each patient. Therefore, is not possible for us to comment or analyze anything regarding to Covid treatment.
R2: I didn’t ask to perform additional analysis just discuss in the discussion section
----------------------------------------------------
Additional comments:
Line 69: Please use records instead of registers.
